# Study on Bulk-Surface Transport Separation and Dielectric Polarization of Topological Insulator Bi_1.2_Sb_0.8_Te_0.4_Se_2.6_

**DOI:** 10.3390/molecules29040859

**Published:** 2024-02-15

**Authors:** Yueqian Zheng, Tao Xu, Xuan Wang, Zhi Sun, Bai Han

**Affiliations:** Department of Electric Engineering, Harbin University of Science and Technology, 52 Xuefu Rd., Nangang, Harbin 150080, China; qianqianchris@163.com (Y.Z.); yang2024tao@163.com (T.X.); sunzhimems@163.com (Z.S.); bhan@hrbust.edu.cn (B.H.)

**Keywords:** nanoscience, topological insulator Bi_1.2_Sb_0.8_Te_0.4_Se_2.6_, transport separation, spectroscopy

## Abstract

This study successfully fabricated the quaternary topological insulator thin films of Bi_1.2_Sb_0.8_Te_0.4_Se_2.6_ (BSTS) with a thickness of 25 nm, improving the intrinsic defects in binary topological materials through doping methods and achieving the separation of transport characteristics between the bulk and surface of topological insulator materials by utilizing a comprehensive Physical Properties Measurement System (PPMS) and Terahertz Time-Domain Spectroscopy (THz-TDS) to extract electronic transport information for both bulk and surface states. Additionally, the dielectric polarization behavior of BSTS in the low-frequency (10–10^7^ Hz) and high-frequency (0.5–2.0 THz) ranges was investigated. These research findings provide crucial experimental groundwork and theoretical guidance for the development of novel low-energy electronic devices, spintronic devices, and quantum computing technology based on topological insulators.

## 1. Introduction

The strong spin–orbit coupling effects and time-reversal symmetry in topological insulators (TIs) give rise to a novel quantum state known as the topological surface state (TSS) [1,2]. TSS emerges as an electronic state inherent to the material’s internal topological properties [3]. These states reside on the surface of topological insulators while exhibiting insulating properties within [4]. Within the TSS, the spin direction of electrons is intricately linked to their momentum direction, a phenomenon termed spin-momentum locking [5,6]. The exceptional nature of these states presents new possibilities for designing novel electronic, spintronic, and quantum devices [7,8,9].

Given the critical role of topological surface states in both theory and applications, there is a desire to observe and obtain accurate electronic transport information regarding TSS. However, in topological insulators, the electronic transport properties of bulk states and surface states are typically coupled, posing challenges for experimental studies [10]. Ideally, researchers aim to individually investigate the electronic transport characteristics of surface and bulk states, but practically, achieving such separation is challenging. The electronic conduction of bulk states often masks the unique properties of surface states. This study aims to mitigate the bulk-surface coupling and extract the transport characteristics by using two approaches: firstly, reducing the material’s defect density to decrease bulk state conduction, thereby amplifying the contribution of surface states; and secondly, employing both electrical and optical testing methods to disentangle and extract information on bulk and surface state transport [11]. To achieve this, quaternary topological insulator thin films of Bi_1.2_Sb_0.8_Te_0.4_Se_2.6_ (BSTS) were fabricated using doping methods to reduce material defect density, enhance bulk state insulation, and accentuate the TSS [12]. The comprehensive Physical Properties Measurement System (PPMS) and Terahertz Time-Domain Spectroscopy (THz-TDS) were utilized to extract information on bulk state transport and surface state transport, respectively [7]. Particularly, in optical testing, these non-contact measurements accurately captured the transport information of TSS. Additionally, considering the complex electromagnetic environments when topological insulators are used as chip or device substrates, the dielectric constant of TIs plays a crucial role in modulating coupling, interference, and isolation among spin quantum bits, while their dielectric loss significantly impacts the efficiency and distance of quantum information transmission. Therefore, we investigated the dielectric response of TIs in the low-frequency (10–10^7^ Hz) and high-frequency (0.5–2.0 THz) ranges [13]. This study provides opportunities to explore phenomena like the spin Hall effect and topological phase transitions, allowing a deeper understanding and utilization of these unique quantum states. It holds significant importance for the development of novel electrons, spintronic devices, and quantum computing technologies based on topological insulators.

In this study, we successfully fabricated Bi_1.2_Sb_0.8_Te_0.4_Se_2.6_ (BSTS) thin films using the Laser Molecular Beam Epitaxy (LMBE) system [14,15]. The process involved a multielement doping method to compensate for two opposing types of charged defects: Te antisite defects and Se vacancy defects. This doping strategy effectively reduced the bulk state conductivity and dissipative pathways in bulk state transport for the thin films. Through this refined material fabrication technique, we achieved a more pronounced presence of TSS in the TI material. The preparation principle, as depicted in Figure 1a, involved the pulsed laser irradiation generated by excimer lasers directed onto the target surface, combined with mixed-target assembly techniques (Figure 1b), resulting in the sputtering of a high-energy plasma beam onto the opposing substrate to facilitate thin film growth. Detailed experimental procedures will be outlined in the Section 4.

## 2. Results

To ensure the reliability of BSTS thin film quality, various testing techniques were employed for molecular structure analysis, morphological and thickness characterization, elemental composition, and band structure analysis of BSTS samples. The results demonstrated that BSTS thin films prepared via LMBE technology exhibited excellent structural quality, laying the foundation for further investigation into their physical properties. The specific findings are as follows.

### 2.1. X-ray Diffraction (XRD): Molecular Structure Analysis

In the XRD analysis, a Japanese Ultima IV Rigaku instrument was utilized, scanning angles ranging from 5° to 90° at a rate of 5°/min. As depicted in Figure 1c, a series of distinct diffraction peaks were observed in the diffractogram, indicating the high crystallinity of the film. These peaks aligned well with the peaks of standard topological insulators, Bi_2_Te_3_, Bi_2_Se_3_, and Sb_2_Te_3_. Furthermore, there were no apparent signs of impurity phases or non-topological insulator phases within the film, indicating that the quaternary doping did not introduce any structural defects. Only the (0 0 3n) peak was present in the film, with the (0 0 9) peak suppressed due to strong substrate peaks, suggesting a preference for the film’s alignment along the c-axis of the silicon substrate. The distinct peak shapes and small full width at half maximum (FWHM) values further confirmed the excellent crystalline quality of the sample. Figure 1d illustrates the hexagonal crystal structure of BSTS, characterized by a layered growth pattern. In this structure, the film stacked along the Z-axis direction, forming a unit of five atomic layers known as the Quintuple Layer (QL) [16]. These QLs exhibited a specific arrangement of A(1)-B-A(2)-B-A(1), where A(1) and A(2) represented two different lattice positions. The A(1) position may be occupied by Se or Te elements, with Se predominantly occupying the central layer of the A(2) position due to its higher electronegativity than Te. The B position was primarily occupied by Bi or Sb elements. Within each single layer, atoms formed strong covalent bonds, while weaker coupling existed between adjacent QLs through van der Waals forces. Precise measurements and calculations determined the lattice constants of BSTS, with lengths of both the a and b axes at 4.23 Å, and the *c*-axis at 29.43 Å.

### 2.2. Scanning Electron Microscopy (SEM) and Atomic Force Microscopy (AFM): Morphology and Thickness Characterization

In SEM analysis, the surface morphology of the BSTS thin film at different magnifications is depicted in Figure 2a,b. Macroscopically, the BSTS film resembled a silver mirror, exhibiting a flat and dense surface without pores or defects when observed under the electron microscope [17]. Surface protrusions appeared as truncated triangular or hexagonal shapes, consistent with the BSTS lattice structure. Observing the cross-section of the sample via AFM revealed a thickness of 25 nm. The average roughness value across a 5000 nm × 5000 nm scan area was approximately 1.29 nm. The thickness of the BSTS quintuple layer measured around 1 nm, indicating a remarkably flat nature of the thin film sample. The cross-sectional morphology and height profile of the sample are depicted in Figure 2c,d.

### 2.3. Energy Dispersive Spectroscopy (EDS): Elemental Composition Analysis

The EDS testing results revealed that the prepared BSTS sample maintained a highly stable stoichiometric ratio (as shown in Table 1), with the elemental spectra presented in Appendix A. In this sample, the total ratio of Bi and Sb atoms to Te and Se atoms consistently maintained a 2:3 ratio, aligning with the molecular structure resembling that of binary topological insulators, as indicated by the earlier XRD analysis. This precise chemical composition provided BSTS material with a more stable topological surface state, which is crucial for its electronic properties. Based on the detailed data from the EDS testing, we ultimately determined the exact chemical formula of the BSTS film to be Bi_1.2_Sb_0.8_Te_0.4_Se_2.6_, accurately reflecting its unique composition and structural characteristics.

### 2.4. Angle-Resolved Photoemission Spectroscopy (ARPES): Band Structure Analysis

In our study, we employed ARPES, a high-precision technique, to conduct a detailed observation and analysis of the electronic band structure of the BSTS thin film [18]. This technique allows for the direct capture of precise information regarding the electronic states on the film surface, including their energy distribution and momentum characteristics, providing crucial evidence for a deeper understanding of the topological properties of the BSTS film. As shown in Appendix A, focused observation of the electronic states near the Fermi level (EF) revealed a linear dispersion of surface states near the Fermi level, depicting a linear relationship between energy and momentum. This resulted in a characteristic band structure known as the “Dirac cone”, which serves as a clear indication of BSTS’s well-defined topological surface state [19]. Additionally, through further meticulous analysis, we determined the position of the Bulk Conduction Band (BCB) approximately 140 meV above the Dirac Point (DP), confirming the existence of a bandgap. In the bulk state, the topological insulator, akin to a regular insulator, exhibited a complete bandgap, indicating the absence of available electronic states near the Fermi level. However, at the surface or edge, conductive surface states were evident, arising from the distinct topological properties between the bulk and surface.

## 3. Discussion

### 3.1. The Extraction of Bulk State Transport Characteristics

In our experiments, we successfully fabricated BSTS thin film samples with a thickness of 25 nm and conducted an in-depth study of their bulk state electron transport characteristics using the Physical Property Measurement System (PPMS). In topological insulators (TIs), the coupling effect between bulk and surface states, particularly pronounced in thinner TI films, often obscures the surface state information, to some extent, by the bulk state features. In PPMS testing, due to the use of contact electrodes, this measurement method inevitably leads to current passing through both the interior and surface of the sample [20]. Consequently, the obtained electronic transport information represented a combined result of contributions from both bulk and surface, as depicted in Figure 3a, which illustrates the dual-channel, bulk-surface transport principle in TI materials and the principle of PPMS measurement [21]. With dissipation channels existing within the bulk, the contribution of the bulk state held a relatively larger proportion in the overall transport characteristics. This dominance of the bulk state becomes more evident with increasing temperatures. Therefore, the results from this PPMS test primarily reflected the bulk state transport characteristics of BSTS material. Resistivity measurements were conducted using the standard four-probe method, while Hall resistance measurements employed the van der Pauw method. By contrasting the bulk state transport results with subsequent surface state transport experiments, a clearer distinction between the contributions of the bulk and surface states could be achieved, thus providing crucial insights into the understanding of the comprehensive electronic properties of BSTS material.

#### 3.1.1. The Current–Voltage (I-V) Characteristics Measurement

In this study, we measured the current–voltage (I-V) characteristics of the sample at room temperature and compared them with the I-V curve of the binary compound Bi_2_Se_3_. As depicted in Figure 3b, the slope of the BSTS I-V curve was relatively lower. This phenomenon suggested that the doping structure in BSTS effectively reduced defects within the sample, such as antistites and vacancies, thereby minimizing electron scattering to the fullest extent. This reduction in electron scattering further decreased the bulk conductivity, significantly enhancing the material’s bulk insulation properties.

#### 3.1.2. The Temperature-Dependent Resistance Measurement/Test of Resistivity Change with Temperature

TI materials exhibit unique conductive behavior where the bulk remains in an insulating state with a bandgap, while the surface showcases gapless conducting characteristics attributed to its topological band structure. In our experiments, by testing the conductivity variation of BSTS samples at different temperatures, we established the relationship between bulk conductivity and temperature (as illustrated in Figure 3c). At low temperatures, bulk conduction weakened due to the suppression of bulk carriers by thermal excitation mechanisms, leading to the dominance of surface conduction. The film’s conductivity slightly decreased, indicating an insulating ground state where electronic conduction was primarily contributed by surface states. BSTS demonstrated a temperature dependence similar to Bi_2_Se_3_ at low temperatures, where the bulk conductivity rapidly decreased with increasing temperature due to the interaction of phonon interactions and impurities, as well as the scattering of Dirac electrons, leading to a “topological breakdown”. Simultaneously, BSTS samples exhibited lower bulk conductivity across different temperature ranges compared to pure Bi_2_Se_3_.

#### 3.1.3. Hall Resistance Variation with Magnetic Field Testing

Through analyzing the relationship between Hall resistance and magnetic field variation (as depicted in Figure 3d), we calculated the carrier concentration and mobility of the samples, as detailed in Table 2. In both semiconductor materials, Bi_2_Se_3_ and Bi_1.2_Sb_0.8_Te_0.4_Se_2.6_ (BSTS), there was similarity in the carrier concentration. However, the mobility of BSTS was more than an order of magnitude lower than that of Bi_2_Se_3_. This significant decrease in mobility in BSTS may be attributed to the compensation of different types of defects, consequently enhancing the material’s bulk insulation performance.

#### 3.1.4. Weak Antilocalization Effect (WAL)

The metallic conduction behavior of Topological Surface States (TSS) primarily arises from the strong spin–orbit coupling at the surface, enabling electrons to move freely along semi-circular and circular trajectories and possess opposite spins in opposite directions. This effect shields surface electrons from impurity influence and backward scattering, explaining the conductivity at the surface and the insulation within the bulk of TI materials. To directly validate the presence of this spin–orbit coupling and its influence on the topological nature of TI materials, we conducted low-temperature magnetic transport experiments, testing the magnetoresistance variation of BSTS samples within a range of −9T to 9T at 2K (as depicted in Figure 3e). We observed sharp peaks in the magnetoresistance of BSTS and Bi_2_Se_3_ near zero magnetic field, which was indicative of the Weak Antilocalization Effect (WAL), a form of quantum interference phenomenon, demonstrating the existence of TSS. WAL is a quantum response in TIs towards classical conductivity, confirming the gapless surface states and helical Dirac cones that are characteristic of three-dimensional topological insulators. Within the −0.5 to 0.5T magnetic field range, BSTS exhibited more pronounced peaks in its magnetoresistance curve, indicating a significantly enhanced WAL effect, thereby affirming BSTS’s superior TSS and stronger spin–orbit coupling compared to binary compounds.

### 3.2. The Extraction of Surface State Transport Characteristics

The unique topological features of TIs primarily resided within their surface layers, typically 1–3 nanometers thick. To accurately extract transport information from the surface states, we aimed to avoid the current penetration issue inherent in conventional contact-based measurements, which could lead to the surface state information being obscured by the bulk transport data from within the material. Thus, we employed a non-contact measurement technique to minimize potential errors introduced by the testing system itself. The terahertz (THz) frequency band, ranging from 0.1 to 10 THz, has garnered attention due to its low single-photon energy, excellent penetration capability, short wavelengths, and rich spectral information. Leveraging terahertz time-domain spectroscopy (THz-TDS), we meticulously measured the optical properties and surface transport information of BSTS samples. As a non-contact measurement technique, THz-TDS effectively reduced errors associated with traditional contact-based electrode testing and mitigated interference caused by the coupling of bulk transport information. Furthermore, the low-energy photons utilized by THz-TDS did not induce transitions in the material’s internal electrons, making it an ideal tool for analyzing the optical properties and surface transport information of ultra-thin nanomaterials. Figure 4a illustrates the fundamental working principle of THz-TDS.

We extracted surface transport information and dielectric parameters of BSTS by recording the temporal distribution of two terahertz pulses, one on a silicon substrate sample and the other on a sample containing BSTS. As depicted in Figure 4b, the temporal evolution of the terahertz pulse electric field in both measurements illustrated a significant shift in the time domain signals due to the presence of BSTS compared to the bare silicon substrate. Employing fast Fourier transform (FFT) on the time-domain waveforms, we obtained the transmission spectrum of the TI-TI sample in the frequency range of 0.5 to 2.0 THz, as shown in Figure 4c. We defined the input THz pulse (no sample) and the output THz pulse (with sample) in the frequency domain as $F_{ref}(\omega )$ and $F_{sam}(\omega )$. The transmission rate T(ω) of the TI-TI sample could be derived from the following equation:(1)$$T(\omega )=\frac{F_{sam}(\omega )}{F_{ref}(\omega )}=A(\omega )\exp\left[i\varphi (\omega )\right]$$

Thereinto, $A(\omega )=\exp\left[-\kappa (\omega )\omega L/c\right]$ is the ratio of magnitude, and φ(ω)=2πn(ω)L/λ is the phase difference of the transmission coefficient at angular frequency ω. We observed a pronounced absorption of terahertz waves by BSTS around 1.5 THz. The phase variation is illustrated in Figure 4d, displaying phase difference data with absolute values less than π. This factor is crucial to the accurate characterization of the sample’s optical properties. When the absolute value of the phase difference exceeds π, the resulting phase spectrum may contain discontinuities that lead to an incorrect representation of optical characteristics. n(ω) is the real part of the complex index of refraction N(ω)=n(ω)+ik(ω), k(ω) is the imaginary part, and λ and L are the free-space wavelength and BSTS film thickness, respectively. According to Equation (1), the real part n(ω) and imaginary part k(ω) of the complex refraction index can be acquired. The complex conductivity function of BSTS is given by the following equation:(2)$$\sigma (\omega )=\sigma_{real}(\omega )+i\sigma_{imag}(\omega )$$

Hence, the real conductivity could be obtained from $\sigma_{real}=2\omega n(\omega )\kappa (\omega )\varepsilon_{0}$ and the imaginary conductivity from $\sigma_{imag}=\omega \varepsilon_{0}\left[\varepsilon_{\infty }-\left(n{\left(\omega \right)}^{2}-\kappa {\left(\omega \right)}^{2}\right)\right]$, where $\varepsilon_{0}$ is the vacuum permittivity, and $\varepsilon_{\infty }$ is the high-frequency dielectric constant of the material.

The results of the BSTS complex conductivity are depicted in Figure 4e. The BSTS film with a thickness of 25 nm exhibited a notable response in surface transport, demonstrating a pronounced increase in conductivity with increasing frequencies. At 1.5 THz, there existed a phonon mode. Due to the low scattering of bulk carriers, the material manifests enhanced metallic behavior at the surface, indicating the dominance of electron-phonon interactions. The surface conductivity of the BSTS sample reached approximately 10^6^ S/m, which was two to three orders of magnitude higher than the bulk conductivity measured via PPMS. This confirmed the successful extraction of surface state transport data through THz-TDS measurement.

### 3.3. The Dielectric Polarization Response of BSTS

Due to its distinctive surface states and electron transport mechanisms, BSTS is considered an ideal material for future spin-electronic devices or quantum computing chips. It is noteworthy that the dielectric constant of BSTS plays a critical role in modulating coupling, interference, and isolation between spin quantum bits, while its dielectric loss significantly affects the efficiency and distance of quantum information transmission. Hence, comprehensive measurement and analysis of the dielectric polarization behavior of topological insulators are particularly crucial for their practical applications. Building upon the successful isolation of bulk-surface transport information within BSTS, we utilized a broadband dielectric spectrometer and THz-TDS to extract the dielectric polarization information of BSTS samples at room temperature in the low-frequency range (10–10^7^ Hz) and high-frequency range (0.5–2.0 THz) through electrical and optical tests in this section. The testing principle of broadband dielectric spectroscopy was illustrated in Appendix A, determining the dielectric constant and the dielectric loss factor by measuring the sample’s capacitance C and resistance R: $\varepsilon^{\prime }=\frac{Cd}{\varepsilon_{0}S}$, $\varepsilon^{\prime \prime }=\frac{d}{\omega \varepsilon_{0}RS}$, $\tan\delta =\frac{\varepsilon^{\prime \prime }}{\varepsilon^{\prime }}$. Here, $\varepsilon^{\prime }$ and $\varepsilon^{\prime \prime }$ represent the real and imaginary parts of the sample’s dielectric constant, respectively. tanδ is the dielectric loss factor, d stands for the sample thickness, and S is the contact area between the sample and electrodes. Integrating the description of surface state transport characteristics and utilizing the complex dielectric conductivity obtained through THz-TDS testing, we derived the complex dielectric constant ε(ω) and dielectric loss factor of the sample: $\varepsilon (\omega )=\varepsilon^{\prime }(\omega )+i\varepsilon^{\prime \prime }(\omega )=\varepsilon_{l}(\omega )+\frac{i\sigma (\omega )}{\omega \varepsilon_{0}}$, where $\varepsilon_{l}$ is the lattice component of the dielectric function. Here, $\varepsilon_{l}$ can be approximated as the low-frequency (static) dielectric constant of BSTS material.

We presented the frequency spectrum of the dielectric constant and dielectric loss of BSTS at room temperature. Figure 5a clearly illustrates that within the frequency range of 10 to 10^3^ Hz, various polarization responses in BSTS, including electronic polarization, dipole reorientation polarization, and relaxation polarization, closely followed the frequency changes, resulting in a complex dielectric response. In the frequency range of 10^3^ to 10^5^ Hz, as the frequency gradually increases, the internal dipole reorientation polarization in BSTS started to lag behind the frequency changes in the external electric field, leading to a hysteresis effect, causing the dielectric constant to decrease with increasing frequency. This phenomenon reflected the lag effect of relaxation polarization, which gradually becomes significant during the frequency variation. Two quantum layers (QL) in BSTS could form a set of dipoles, which could adjust their dipole moment direction corresponding to the polarization changes in the external electric field. The distribution of positive and negative charges in these dipoles contributed to reducing the influence of the internal electric field. Upon entering the frequency range of 10^5^ to 10^7^ Hz, where relaxation polarization fails to be established, BSTS polarization was primarily contributed by displacement polarization, leading to a tendency of the dielectric constant toward a stable value.

In Figure 5b, we observed a decreasing trend in tanδ as the frequency increased within the range of 10 to 10^3^ Hz. This decrease was attributed to the dominance of conduction losses at lower frequencies, where relaxation polarization losses approached zero. As the frequency raised, conduction losses gradually diminished. Within the 10^3^ to 10^5^ Hz range, tanδ increased with frequency. This increase was due to the relaxation polarization losses progressively escalating with frequency, leading to amplified dielectric losses in the material, and BSTS exhibited a peak value in dielectric loss within the relaxation region. Within the 10^5^ to 10^7^ Hz frequency range, dielectric losses gradually decreased. This reduction primarily raised from the inability of dipole reorientation polarization to track changes in the external electric field direction at high-frequencies, resulting in decreased relaxation polarization losses and a gradual decline in tanδ for the high-frequency region.

As shown in Figure 5c,d, the real and imaginary parts of the dielectric constant exhibited electronic displacement polarization behavior around 1.5 THz, reflecting the material’s dielectric polarization characteristics. In the high-frequency region, other internal polarization processes failed to keep up with the rapid changes in the electric field. Consequently, these polarization mechanisms exhibited weaker responses to the electric field, leading to reduced dielectric losses. Within the BSTS thin film, the presence of surface states induced polarization behavior different from bulk materials, potentially resulting in unique dielectric responses and polarization characteristics to external electric fields.

## 4. Materials and Methods

LMBE: Film Growth: We utilized the LMBE system (LMBE450 type, Japan) for BSTS film fabrication (Figure 1). Initially, cleaned Si substrates were placed on the substrate holder within the epitaxy chamber. Subsequently, a pulsed laser from the LMBE system, using four inert gases (He, Ne, F, and Kr) as the energy sources, was employed. The laser was precisely directed through adjusted reflectors and transmitters, focusing and bombarding four targets inside the epitaxy chamber (Bi_2_Se_3_, with the target purity of 99.999%, Sb_2_Te_3_, with the target purity of 99.999%, Te, with the target purity of 99.99%, and Se, with the target purity of 99.99%). Simultaneously, as the target holder rotated uniformly, the pulsed laser uniformly illuminated each area of the composite-targets in the epitaxy chamber. This process generated Bi, Se, Sb, and Te atoms, sputtering onto the opposite substrate in the form of a plasma beam and completing the growth of the BSTS film.

PPMS: The testing equipment was the QUANTUM DESIGN PPMS-9. The measurement of resistivity employed the four-probe method. Hall resistivity was determined using the van der Pauw method.

AFM: Non-contact mode is employed, utilizing a diamond (boron-doped) probe, with a scanning area of 5 × 5 μm and a scanning speed of 5 Hz.

SEM: The working distance (WD) is set at 6.5 mm, operating in secondary electron imaging (SEI) mode, with an accelerating voltage of 3.00 kV.

EDS: Scanning mode is configured for surface scanning.

THz-TDS: The experiment utilized the standard transmission type THz-TDS (Terahertz Time-Domain Spectroscopy). The laser source employed was the C-Fiber 780 (Menlo Systems, Martinsried, Germany), generating pulses of 100–120 fs duration, operating at a frequency of 780 ± 10 nm, with a repetition rate of 100 ± 1 MHz and an average power of 100 mW. The optical setup involves a femtosecond laser. One beam of the femtosecond laser, directed by a beam splitter, is used to excite the photoconductive antenna. The pump power is 200mW, and the bias voltage for the radiating antenna is set at 300 V.

## 5. Conclusions

This study successfully fabricated BSTS film with a thickness of 25 nm, compensating for intrinsic defects in binary TI materials through doping techniques. By employing electrical and optical methodologies, the transport properties of BSTS were effectively distinguished between bulk and surface states, indicating insulating behavior in the bulk state while demonstrating notable conductivity in the surface state. Simultaneously, investigations into the dielectric properties of BSTS across low-frequency (10–10^7^ Hz) and high-frequency (0.5–2.0 THz) ranges revealed distinct polarization behaviors at different frequencies. These findings hold significant implications for manipulating the coupling, interference, and isolation among spin quantum bits. Such discoveries provide crucial theoretical and experimental foundations for the utilization of topological insulators to develop novel quantum and low-power electronic devices.

## Figures and Tables

**Figure 1 molecules-29-00859-f001:**
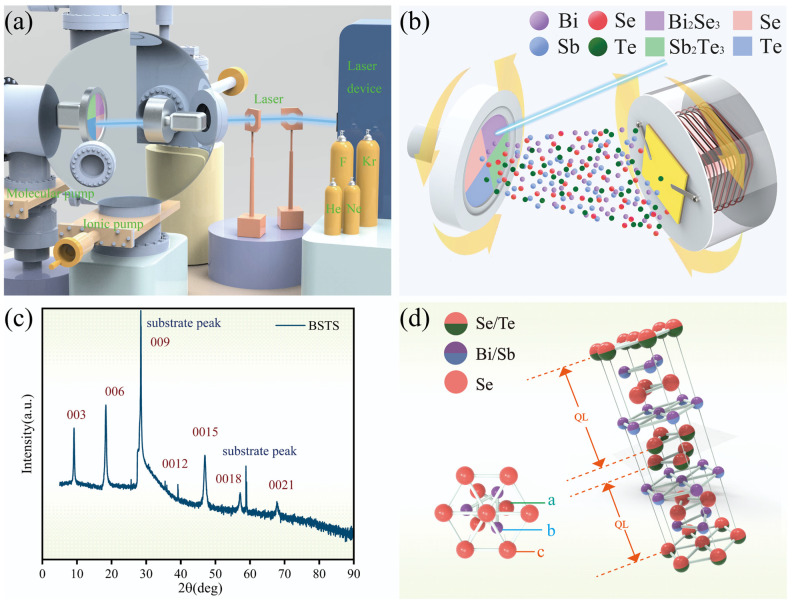
(**a**) Schematic diagram of Laser Molecular Beam Epitaxy (LMBE) equipment preparation. (**b**) Schematic diagram of mixed-target assembly. (**c**) XRD diffraction pattern of the BSTS sample. (**d**) Schematic illustration of the molecular structure of the BSTS sample.

**Figure 2 molecules-29-00859-f002:**
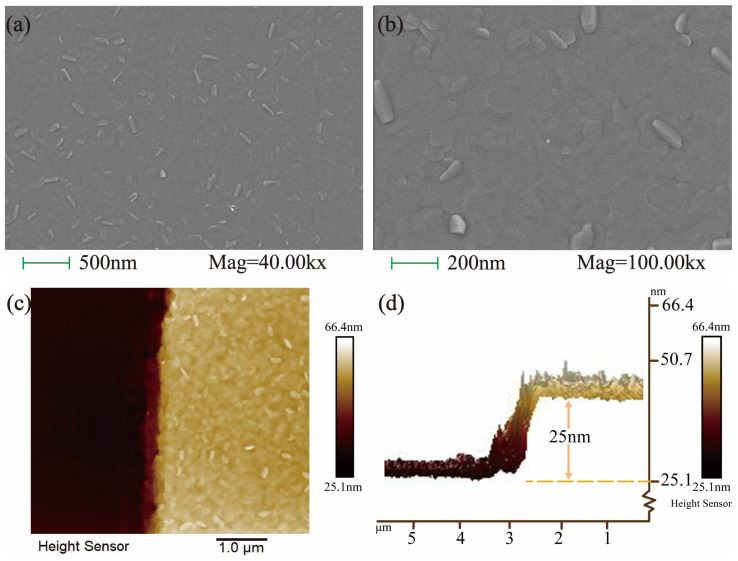
(**a**) Surface morphology of the BSTS sample at 40.00 kx magnification in SEM characterization. (**b**) Surface morphology of the BSTS sample at 100.00 kx magnification in SEM characterization. (**c**) Cross-sectional morphology of the BSTS sample in AFM characterization. (**d**) Cross-sectional height profile of the BSTS sample, demonstrating a film thickness of 25 nm.

**Figure 3 molecules-29-00859-f003:**
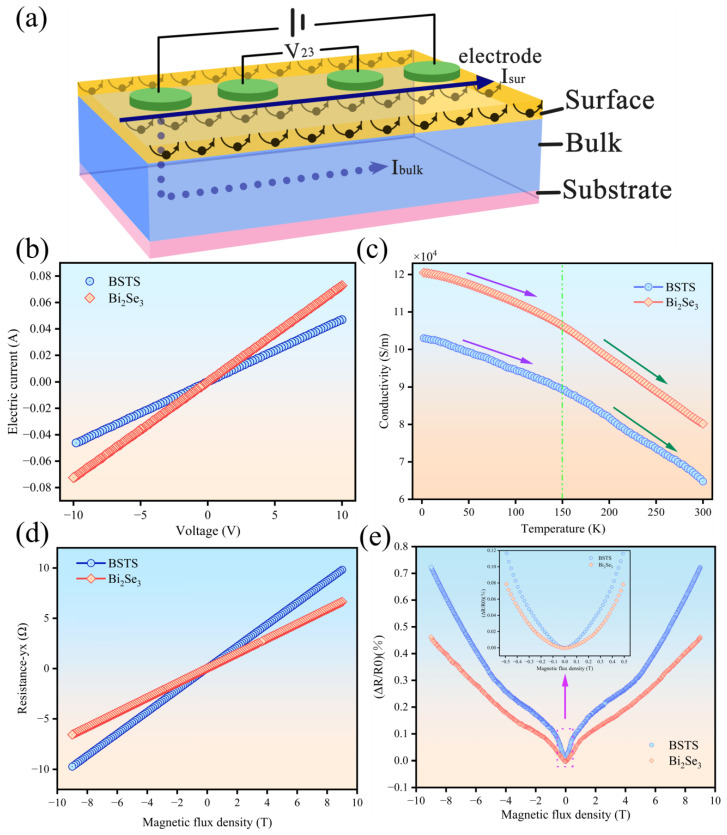
(**a**) Schematic illustration of the dual-channel, bulk-surface transport in the BSTS and Bi_2_Se_3_ sample and the principle of PPMS measurement. (**b**) Current–voltage (I-V) relationship of the BSTS and Bi_2_Se_3_ sample at room temperature. (**c**) Relationship between bulk conductivity and temperature in the BSTS and Bi_2_Se_3_ sample. (**d**) Relationship between Hall resistance and magnetic field variation in the BSTS and Bi_2_Se_3_ sample. (**e**) Magnetoresistance variation of the BSTS and Bi_2_Se_3_ sample with magnetic field changes in the range of −9T to 9T at 2K temperature.

**Figure 4 molecules-29-00859-f004:**
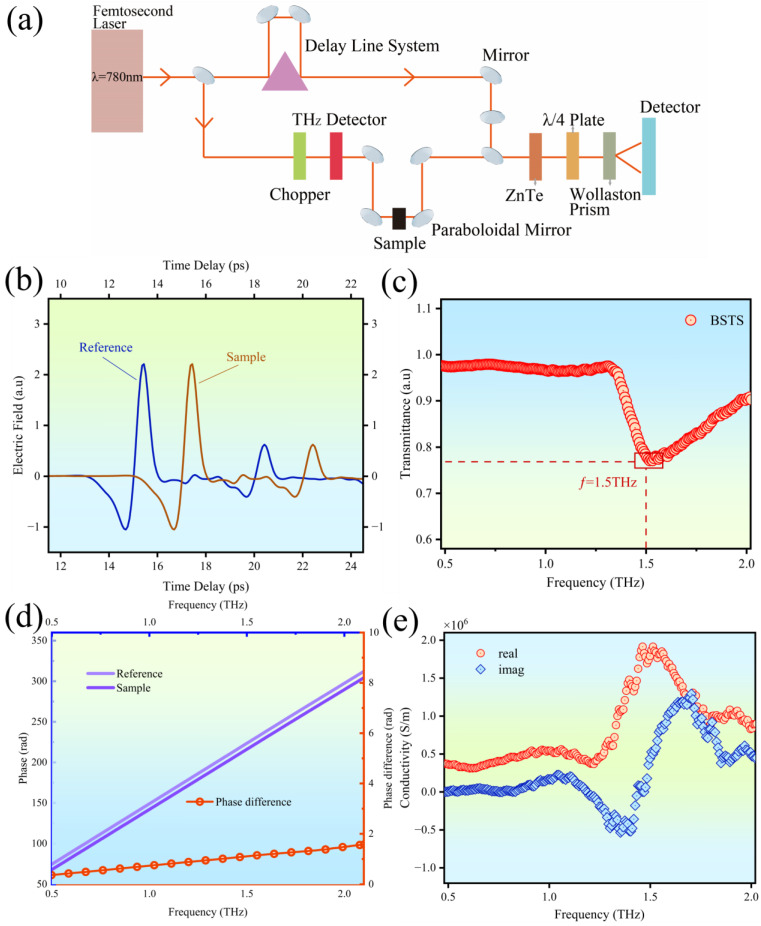
(**a**) Fundamental working principle of Terahertz Time-Domain Spectroscopy (THz-TDS). (**b**) Time-domain distribution of the Terahertz pulses for the BSTS sample and reference sample. (**c**) Terahertz transmission spectrum of the BSTS sample. (**d**) Phase variation of the BSTS sample in the frequency domain. (**e**) Real and imaginary parts of surface conductivity of the BSTS sample.

**Figure 5 molecules-29-00859-f005:**
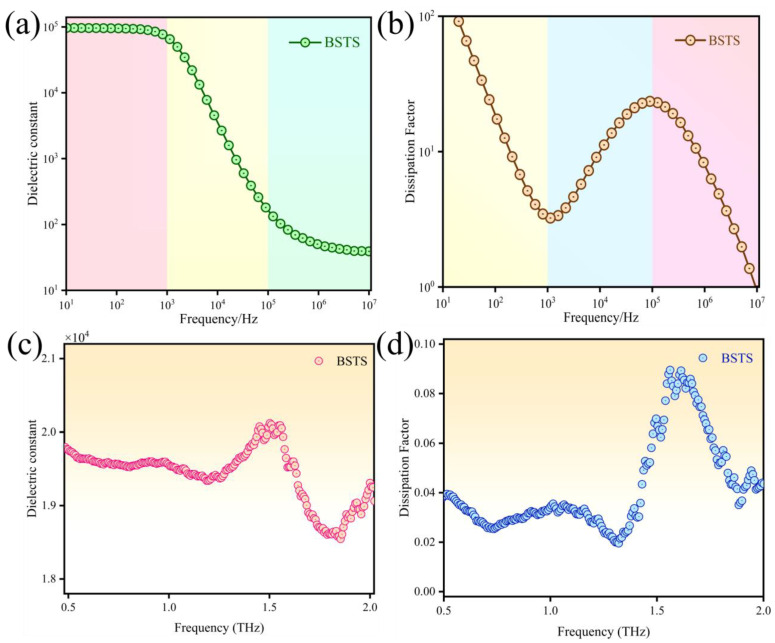
(**a**) Curve showing the variation in dielectric constant of BSTS sample within the frequency range of 10–10^7^ Hz. (**b**) Curve depicting the change in loss factor of the BSTS sample within the frequency range of 10–10^7^ Hz. (**c**) Curve illustrating the fluctuation in dielectric constant of the BSTS sample within the frequency range of 0.5–2.5THz. (**d**) Curve displaying the alteration in loss factor of the BSTS sample within the frequency range of 0.5–2.5 THz.

**Table 1 molecules-29-00859-t001:** Atomic Weight and Percentage Composition of the Four Elements in BSTS.

Element	Weight Percentage	Atomic Percentage
Bi	37.76	21.05
Te	9.86	9.00
Sb	14.13	13.52
Se	38.25	56.43
Total	100.00	100.00

**Table 2 molecules-29-00859-t002:** BSTS Electrical Test Date.

Sample	Temperature	Carrier Areal Density	Carrier Bulk Density	Hall Coefficient	Bulk Carrier Mobility
BSTS	300 K	5.74 × 10^14^ cm^−2^	2.87 × 10^19^ cm^−3^	0.022 cm^3^/C	14.14 cm^2^/Vs
Bi_2_Se_3_	300 K	1.92 × 10^14^ cm^−2^	7.67 × 10^19^ cm^−3^	0.032 cm^3^/C	653.5 cm^2^/Vs

## Data Availability

The data that supports the findings of this study are available within the article and Appendix A.

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
