# Peer review of "Study on Bulk-Surface Transport Separation and Dielectric Polarization of Topological Insulator Bi_1.2_Sb_0.8_Te_0.4_Se_2.6"

_molecules, 2024, doi:10.3390/molecules29040859_

Round 1

Reviewer 1 Report

Comments and Suggestions for Authors

The article "Study on Bulk-Surface Transport Separation and Dielectric Polarization of Topological Insulator Bi1.2Sb0.8Te0.4Se2.6" focuses on the fabrication and analysis of quaternary topological insulator thin films. It explores the separation of transport characteristics between the bulk and surface of topological insulator materials using advanced techniques like Physical Properties Measurement System (PPMS) and Terahertz Time-Domain Spectroscopy (THz-TDS). The study also investigates the dielectric polarization behavior of these materials in both low and high-frequency ranges, aiming to provide experimental and theoretical insights for developing new low-energy electronic, spintronic devices, and quantum computing technology based on topological insulators.

Strengths:

  1. Innovative and Significant Topic: The article addresses a crucial and contemporary topic in the field of topological insulators, contributing to the advancement of electronic and spintronic technologies.
  2. Methodology: The use of advanced experimental techniques, such as terahertz time-domain spectroscopy and physical property measurement systems, provides a robust and comprehensive approach to material property analysis.
  3. Presentation of Results: Clear presentation of results, including detailed observations and band structure analysis, essential for understanding the properties of topological insulators.

Weaknesses:

  1. Data Accessibility: The lack of access to raw measurement data may limit the ability of other researchers to verify and reproduce the results.
  2. Research Limitations: The article could have discussed potential methodological limitations and their impact on the conclusions more thoroughly.
  3. Theoretical Detail: While the experimental results are detailed, the article could benefit from a more extensive theoretical discussion on the underlying mechanisms of the observed phenomena.

I would recommend sending the article for a minor review.

Author Response

Dear reviewer:

Thanks to the reviewers for their detailed comments and valuable suggestions. As mentioned in your review, this paper aims to use advanced detection techniques to accurately extract the surface-in-body electrical transport characteristics of topological insulators to provide theoretical support for their applications in quantum computing and low-energy spin devices. At the same time, based on your comments, we have also found shortcomings in the article. Below, we will reply and modify according to your three suggestions.

Thank you again for your time and consideration of our manuscript.

Sincerely,

Xuan Wang

Questions and Answers:

  1. Data Accessibility: The lack of access to raw measurement data may limit the ability of other researchers to verify and reproduce the results.”

Answer: The research in this paper is original, we use LMBE system to prepare a new doping sample BSTS, and use a variety of test methods to demonstrate, using THz-TDS and PPMS system to extract data on the electrical characteristics of the body surface. All the testing and data extraction work was done independently by the experimental group. Therefore, we also mentioned in the system that we will provide all the detailed raw data for researchers who need it. Some of the data that can be presented have been provided in the body table and supplementary file. We hereby pledge to actively cooperate with you and your journal to provide primary data for researchers to conduct research.

  1. Research Limitations: The article could have discussed potential methodological limitations and their impact on the conclusions more thoroughly.”

Answer: Thank you to the reviewers for their comments on the methodological limitations of the paper; it is evident that you have dedicated a significant amount of effort to the peer-review process. We express our gratitude. Below, we will elaborate on two main aspects: preparation and testing.:

Preparation: We achieved controlled growth of BSTS samples using the LMBE system, employing an innovative mixed-target technique to simplify the preparation process and address element deficiency issues. This is detailed more extensively in our previously published articles. (https://doi.org/10.1016/j.jmrt.2022.05.097) However, it must be acknowledged that this preparation approach has its limitations. For instance, the relatively slow growth rate of the LMBE system, coupled with the need to control multiple variables during the preparation process, hinders the efficient production of large-sized samples. Additionally, in-situ annealing under high vacuum conditions is time-consuming, reducing preparation efficiency. Furthermore, the unique characteristics of topological insulators manifest predominantly in their conductive surface states, imposing stringent requirements on the surface quality of samples. For instance, ARPES testing delves into information within a few nanometers of the sample surface. Our samples, when exposed to the air, are prone to oxidation and contamination. Hence, in many tests, we need to apply a protective layer (such as Se) to the samples. Subsequently, the layer is heated and removed in the testing system, adding complexity to the workflow and introducing the possibility of errors. This challenge is unavoidable for standalone LMBE systems. The optimal solution involves integrating the surface preparation equipment and testing equipment with the LMBE system, forming a closed system where samples can seamlessly transition between systems. However, this imposes stringent requirements on laboratory conditions, and the interconnection of large-scale equipment is a substantial undertaking.

Bulk Transport Testing: We utilized the PPMS system to extract bulk transport information from BSTS samples. In the experiment, it was necessary to deposit electrodes on the sample surface and apply voltage. Due to the sample's thinness of only 25nm, the electrode deposition inevitably affected the sample surface, introducing errors. Simultaneously, as mentioned in the article (lines 143 to 145 of the original article), we faced the issue of a dual-channel effect, where the information obtained reflected the combined contributions of both surface and bulk. In this particular experiment, the bulk contribution nearly overshadowed the surface state contribution, though we cannot deny its existence. This introduces an unavoidable error in extracting bulk transport information. To address this, we will continue to explore alternative testing methods in future work to minimize such inaccuracies.

  1. Theoretical Detail: While the experimental results are detailed, the article could benefit from a more extensive theoretical discussion on the underlying mechanisms of the observed phenomena.”

Answer: Thank you for the valuable suggestions from the reviewers. The research presented in this paper builds upon our previous study results. The need to find a suitable method for extracting the transport information of topological insulator surface and bulk arises from its unique physical characteristics: exhibiting surface conduction states while maintaining insulating states in the bulk. This distinctive physical mechanism stems from the unique band structure and spin-orbit coupling effects of topological insulators. Spin-orbit coupling is a phenomenon where electron spin interacts with its orbital motion, and on the surface of topological insulators, this coupling can lead to unique surface states. Topological insulators typically manifest topologically protected surface states induced by spin-orbit coupling, showcasing special properties in electronic transport, such as the edge states or surface states of topological insulators. These unique surface states enable widespread applications of topological insulators in various fields. Therefore, this paper delves deeper into the extraction of these surface states, building upon our previous work where we introduced the distinctive transport mechanism itself. (https://doi.org/10.1016/j.apsusc.2023.158052)

Reviewer 2 Report

Comments and Suggestions for Authors

In this manuscript, Zheng et al fabricated a topological insulator thin film of BSTS and measured its electron transport properties in the bulk and at the surface. The experimental results are clearly presented and the manuscript is well written. I would recommend it for publication after the following minor concerns are addressed:

1. The THz-TDS measurements were carried out in a transmission geometry. In this case, how was the surface response separated from the bulk contribution?

2. Some experimental details are missing in the Methods section. For instance, SEM, AFM, EDS, etc. Also, the pulse energies in the THz setup should be mentioned.

3. In Fig. 3b and 3d, the BSTS curves are in blue, while in 3c and 3e, they are in red. Consider making the colors consistent in different panels.

4. Curves of Bi2Se3 are shown in Fig. 3b-d but are not mentioned in the caption.

Author Response

Dear reviewer:

Thank you for the contributions and efforts made by the reviewers in reviewing this paper. We also appreciate your affirmation of our work. Your valuable suggestions will aid us in refining the manuscript and have provided clear direction for our future work. Below, we will address the four questions you have raised.

Thank you again for your time and consideration of our manuscript.

Sincerely,

Xuan Wang

Questions and Answers:

  1. “The THz-TDS measurements were carried out in a transmission geometry. In this case, how was the surface response separated from the bulk contribution?”

Answer: Terahertz time-domain spectroscopy (THz-TDS) is typically sensitive to the surface and shallow-layer information of materials. This is because the wavelength of terahertz radiation is relatively long, usually in the range of micrometers and nanometers, resulting in limited penetration depth. When testing samples, the information primarily extracted is from the surface layer and the vicinity of the material's surface that interacts with terahertz radiation. This surface sensitivity can be utilized to investigate the electromagnetic properties, thickness, and surface structure of materials.

Simultaneously, due to the low frequency of terahertz photons, their energy is also relatively low. This low-energy characteristic renders terahertz radiation less penetrating for many materials and avoids inducing electron transitions within the bulk of topological insulators. The unique properties of the terahertz frequency range make it suitable for extracting surface information from our materials. The phenomenon of bulk-surface coupling in topological insulators, primarily induced by the testing method, is significantly mitigated by our non-contact optical measurements. Furthermore, by comparing the internal transport information, we observed a substantial enhancement in surface conductivity by two to three orders of magnitude. With such a significant boost in conductivity, coupled with the surface conduction and bulk insulation properties of topological insulators, we believe we have successfully isolated the surface information from the material.

  1. “Some experimental details are missing in the Methods section. For instance, SEM, AFM, EDS, etc. Also, the pulse energies in the THz setup should be mentioned.”

Answer: Thank you for your suggestions. Below, we would like to provide additional details regarding the issues you raised:

AFM: Non-contact mode is employed, utilizing a diamond (boron-doped) probe, with a scanning area of 5×5 micrometers and a scanning speed of 5Hz.

SEM: The working distance (WD) is set at 6.5mm, operating in secondary electron imaging (SEI) mode, with an accelerating voltage of 3.00kV.

EDS: Scanning mode is configured for surface scanning.

THz-TDS: The optical setup involves a femtosecond laser. One beam of the femtosecond laser, directed by a beam splitter, is used to excite the photoconductive antenna. The pump power is 200mW, and the bias voltage for the radiating antenna is set at 300V.

  1. “In Fig. 3b and 3d, the BSTS curves are in blue, while in 3c and 3e, they are in red. Consider making the colors consistent in different panels.”

Answer: We apologize for the oversight in the creation of this image, and any inconvenience it may have caused. We sincerely regret the error, and we have now uploaded the corrected image for your reference.

  1. “Curves of Bi2Se3 are shown in Fig. 3b-d but are not mentioned in the caption.”

Answer: Thank you for your meticulous review. We have incorporated Bi2Se3 into the caption of the original figure as indicated in the parentheses. (Lines 156 to 160 of the original article)

“Figure 3. (a) Schematic illustration of the dual-channel bulk-surface transport in the BSTS and Bi2Se3 sample and the principle of PPMS measurement. (b) Current-voltage (I-V) relationship of the BSTS and Bi2Se3 sample at room temperature. (c) Relationship between bulk conductivity and temperature in the BSTS and Bi2Se3 sample. (d) Relationship between Hall resistance and magnetic field variation in the BSTS and Bi2Se3 sample. (e) Magnetoresistance variation of the BSTS and Bi2Se3 sample with magnetic field changes in the range of -9T to 9T at 2K temperature.”
